# Boosting the Electrocatalytic Oxygen Reduction Activity of MnN_4_-Doped Graphene by Axial Halogen Ligand Modification

**DOI:** 10.3390/molecules29153517

**Published:** 2024-07-26

**Authors:** Shaoqiang Wei, Ran Zhao, Wenbo Yu, Lei Li, Min Zhang

**Affiliations:** 1College of Physics and Electronic Information, Inner Mongolia Normal University, Hohhot 010022, China; 15547415597@163.com (S.W.); yyxqc369@163.com (R.Z.); 16647129980@163.com (W.Y.); 2Inner Mongolia Key Laboratory for Physics and Chemistry of Functional Materials, Hohhot 010022, China

**Keywords:** axial halogen ligand, oxygen reduction reaction, density functional theory, electronic structure, single-atom catalysts

## Abstract

Exploring highly active electrocatalysts as platinum (Pt) substitutes for the oxygen reduction reaction (ORR) remains a significant challenge. In this work, single Mn embedded nitrogen-doped graphene (MnN_4_) with and without halogen ligands (F, Cl, Br, and I) modifying were systematically investigated by density functional theory (DFT) calculations. The calculated results indicated that these ligands can transform the *d*_yz_ and *d*_xz_ orbitals of Mn atom in MnN_4_ near the Fermi-level into $d_{z^{2}}$ orbital, and shift the *d*-band center away from the Fermi-level to reduce the adsorption capacity for reaction intermediates, thus enhancing the ORR catalytic activity of MnN_4_. Notably, Br and I modified MnN_4_ respectively with the lowest overpotentials of 0.41 and 0.39 V, possess superior ORR catalytic activity. This work is helpful for comprehensively understanding the ligand modification mechanism of single-atom catalysts and develops highly active ORR electrocatalysts.

## 1. Introduction

With the ever-increasing energy crisis and environmental issues, a major challenge facing mankind is to develop eco-friendly and renewable energy conversion and storage devices, such as fuel cells and rechargeable batteries [1,2,3]. However, due to the sluggish kinetics of the cathodic oxygen reduction reaction (ORR), these devices are heavily dependent on Pt catalysts to accelerate the reaction process [4,5]. Unfortunately, the natural scarcity and prohibitive cost of Pt greatly increased the large-scale commercialization cost of fuel cells and rechargeable batteries [6,7]. Therefore, it is imperative to explore inexpensive and high-performance materials as alternatives to Pt catalysts.

In this case, graphene-supported nitrogen-doped transition metal (M−N−G) single atom catalysts (SACs) have attracted increasing attention because of their efficient use of atoms and their outstanding mechanical flexibility, and which are considered as promising alternative material to Pt catalysts [8,9,10]. However, their catalytic activities are greatly impeded by the unsatisfactory adsorption strength for ORR intermediates [11,12]. Optimizing the electronic structure of metal active center can further improve their catalytic activities, including axial ligand coordination engineering, heteroatom doping, and the construction of diatomic or polyatomic active sites [13,14,15]. Among which, axial ligand coordination engineering was proved to be a novel and feasible strategy to boost the catalytic activity of M−N−Gs, especially for halogen ligands [13,16,17,18,19,20,21]. For example, Han et.al. successfully prepared an FeN_4_Cl catalyst with a half-wave potential of 0.921 V, which showed remarkably enhanced ORR catalytic activity compared to FeN_4_. Furthermore, density functional theory (DFT) calculations indicated that the axial Cl ligand can effectively modulate the adsorption capacity of FeN_4_ for reaction intermediates and transform the rate-determination step (RDS) of *OH reduction into *OOH formation, resulting in the superior ORR activity of FeN_4_Cl [22]. Both experiments and DFT calculations have confirmed that FeN_4_I possesses superior ORR catalytic activity, and the corresponding half-wave potential can reach to 0.948 V, which should be attributed to the appearance of Fe $d_{z^{2}}$  orbital with a lower energy level caused by I modification [23]. Chen et al. studied a series of MN_4_X (M = Fe, Co, and Ni; X = F, Cl, Br, and I) catalysts by performing DFT calculations, and found that axial coordination of X can modulate the electronic structure of M active center to alter the binding of MN_4_ with reaction intermediates. In particular, CoN_4_Cl and CoN_4_Br, respectively, with ultralow overpotentials of 0.25 and 0.26 V possess excellent ORR catalytic activity [24]. In addition, the axial halogen ligand can regulate the selectivity of the ORR pathway. For example, different from the four-electron (4e^−^) ORR pathway on CuN_4_ and ZnN_4_, CuN_4_Cl and ZnN_4_Br are more favorable with a two-electron (2e^−^) ORR pathway to form H_2_O_2_, and the overpotentials are only 0.07 and 0.05 V, respectively [25]. Although certain M−N−Gs modified by halogen ligands have shown excellent catalytic activity, a comprehensive examination of single Mn−embedded and N−doped graphene (MnN_4_) modified by various halogen ligands (F, Cl, Br, and I) is still lacking. According to available studies, the overpotential of ORR on MnN_4_ is about 0.90 V [26], which is significantly higher than those for Pt (100) (0.47 V) [27] and Pt (111) (0.44 V) [28]. Therefore, it is expected to reduce the ORR overpotential and enhance the catalytic activity of MnN_4_ by halogen ligand modification. Moreover, how the halogen ligands regulate the electronic structures, reaction intermediates adsorption characteristics, ORR catalytic activity, and reaction mechanism of MnN_4_ remain to be further explored.

Herein, halogen ligands X (X = F, Cl, Br, and I) were introduced to axially modify the active center of MnN_4_, named MnN_4_−X. The structural stabilities, electronic properties, and catalytic characters of MnN_4_ and MnN_4_−X were investigated by DFT calculations. Our calculated results showed that axial halogen ligands can significantly improve the ORR activity of MnN_4_. In particular, the overpotentials of MnN_4_−Cl, MnN_4_−Br, and MnN_4_−I are 0.44, 0.41, and 0.39 V, respectively, which are comparable to or even better than those of Pt catalysts. This work provides great significance for the design and development of low cost and high efficiency graphene-based electrocatalysts.

## 2. Results and Discussion

### 2.1. Structural Model

A 5 × 5 graphene supercell containing 50 carbon atoms was used to construct a MnN_4_ catalyst, as shown in Figure 1. It can be seen that all of the atoms are kept in the same plane, and the average bond length of Mn−N bonds, the formation energy (*E*_f1_), and the binding energy between Mn and N_4_−G (*E*_b1_) are 1.916 Å, −3.16 eV, and 6.90 eV, respectively. These values are consistent with those (1.920 Å, −3.07 eV, and 6.84 eV) in available studies [26,29,30], implying the reliability of our calculations. Moreover, the negative *E*_f1_ and the *E*_b1_ larger than the cohesive energy (3.87 eV) of bulk Mn indicate the good structural stability of MnN_4_. Hence, both sides of MnN_4_ can adsorb ORR intermediates, potentially realizing an axial ligand coordination with the Mn atom [8,31]. Thus, axial halogen ligand X was introduced into MnN_4_ to form MnN_4_−X catalysts (see Figure 1). Due to the effect of X, the average bond lengths of Mn−N bonds in MnN_4_−F, MnN_4_−Cl, MnN_4_−Br, and MnN_4_−I are, respectively, stretched into 1.975 Å, 1.975 Å, 1.973 Å, and 1.971 Å. Meanwhile, the Mn atom is deviated from the plane of graphene. To evaluate the structural stabilities MnN_4_−X, the binding energies (*E*_b2_) between X and MnN_4_ and the formation energy (*E*_f2_) for MnN_4_−X were calculated by Equations (10) and (12) and are presented in Figure 2. One can find that all MnN_4_−X structures have smaller formation energy than that of MnN_4_, indicating the better structural stability of MnN_4_−X than MnN_4_. Moreover, the positive values of *E*_b2_ suggest the good combinations between X and MnN_4_. Furthermore, the AIMD simulations show that the energy of each MnN_4_ and MnN_4_−X structure smoothly oscillates around a certain value, and there is no bond breakage, although their structures were wrinkled to some extent, as indicated by Appendix A. These results indicate that all structures mentioned above are thermodynamically stable. In addition, according to the calculated positive values of *U*_diss_ in Appendix A, all structures are verified to be electrochemically stable. Therefore, all MnN_4_ and MnN_4_−X structures are thermodynamically and electrochemically stable and are considered in the following discussions. From the Bader charge analyses in Appendix A, the Mn atom loses 1.28 e^−^ to N_4_−G in MnN_4_. After introducing X into MnN_4_, the ligands of F, Cl, Br, and I, respectively, obtained 0.65, 0.59, 0.53, and 0.47 e^−^ from the Mn atom, in agreement with the electronegative order of F (3.98) > Cl (3.16) > Br (2.96) > I (2.66). The Mn atom loses more charges than that in MnN_4_, presenting the order of MnN_4_−F (1.50 e) > MnN_4_−Cl (1.40 e) > MnN_4_−Br (1.36 e) > MnN_4_−I (1.30 e). Therefore, the X ligand modification would lead to different charge densities of Mn atoms and affect the adsorption activities for ORR intermediates.

### 2.2. Adsorption of Intermediates

According to the Sabatier’s principle [32], the catalytic activity of catalysts is strongly dependent on the adsorption strength for reaction intermediates. Thus, the adsorption characteristics of ORR intermediates on MnN_4_ and MnN_4_−X were explored. After optimizing all horizontal and vertical configurations of *O_2_, *OOH, *O, and *OH with different orientations on all catalysts, the most stable adsorption structures were obtained and are presented in Figure 3. It can be seen from this figure that all intermediates are preferably adsorbed on the Mn site through an O atom, indicating that the Mn atom is the active center in MnN_4_ and MnN_4_−X. This is understandable since the net spin charges mainly distribute on the Mn atom in MnN_4_ and MnN_4_−X (see Figure 4). In theory, the adsorption of the O_2_ molecule is regarded as a prerequisite for triggering the ORR, so it is firstly considered. As shown in Figure 3, the O_2_ molecule tends to adsorb on the Mn site through side-on configuration in MnN_4_ but through end-on configuration in MnN_4_−X. The adsorption energies of O_2_ on MnN_4_, MnN_4_−F MnN_4_−Cl, MnN_4_−Br, and MnN_4_−I, respectively, are −1.34, −0.51, −0.25, −0.26, and −0.44 eV (see Appendix A), and MnN_4_ exhibits the strongest adsorption capability to O_2_. Among this, the adsorption energy of O_2_ on MnN_4_ is in agreement with the values of −1.33 and −1.43 eV in previous studies [9,26,29]. Meanwhile, the O−O bonds are, respectively, elongated from 1.232 Å in gas-phase O_2_ to 1.392, 1.302, 1.305, 1.304, and 1.295 Å. Since the spin charges of the active center and the Bader charges analysis can be used to describe the binding strength between the metal center and adsorbate in the catalysts [33,34,35], the spin charges of the Mn atom and the charges transfer between O_2_ and the catalysts were calculated to explore the difference of O_2_ adsorption strength on MnN_4_ and MnN_4_−X. Figure 4 and Appendix A show that the spin charge of the Mn atom in MnN_4_ is significantly higher than that in MnN_4_−X, and the Mn atom in MnN_4_ can transfer more charges (0.70 e^−^) to O_2_ than that in MnN_4_−X (0.50, 0.53, 0.51, and 0.44 e^−^, respectively, for MnN_4_−F, MnN_4_−Cl, MnN_4_−Br, and MnN_4_−I), indicating the strongest interactions between O_2_ and MnN_4_. However, the adsorption energy of MnN_4_ for O_2_ is much stronger than that on Pt (−1.10 eV) [27], FeN_4_ (−1.13 eV), and CoN_4_ (−0.90 eV) [36], which may hinder the subsequent reaction processes. In contrast, MnN_4_−X is likely to possess better ORR activity than Pt, FeN_4_, CoN_4_, and MnN_4_. Similar to the adsorption of the O_2_ molecule, the X ligand can weaken the interactions between *OOH, *O, and *OH and MnN_4_ to different extents, as shown in Appendix A. Therefore, the X ligand modification can effectively alter the adsorption capacity of MnN_4_ towards reaction intermediates, and then optimize the ORR catalytic activity.

### 2.3. Catalytic Performance

According to the 4e^−^ reaction pathway (Equations (1)–(4)) of ORR, the free energy change for all catalysts was calculated by Equation (7) and presented in Appendix A and Figure 5 to evaluate the catalytic performance. At zero electrode potential, the free energy in the whole reaction progress is gradually decreased for all catalysts, which means that each step of the reaction is exothermic and can spontaneously proceed. When increasing electrode potential, MnN_4_, MnN_4_−F, MnN_4_−Cl, MnN_4_−Br, and MnN_4_−I, respectively, present working potentials of 0.34, 0.59, 0.79, 0.82, and 0.84 V for maintaining the exothermic reaction process (see Figure 5). Among this, MnN_4_−Br and MnN_4_−I with larger working potentials are more prone to promote ORR compared with other catalysts. Under the standard ORR potential of 1.23 V, some reaction steps become uphill and are thermodynamically unfavorable. In particular, the hydrogenation of *OH to H_2_O with the maximum free energy barrier is the RDS for the ORR on all catalysts. In order to intuitively compare the catalytic activity, the theoretical overpotential is calculated through Equation (8) and illustrated in Figure 5. Note that the lower overpotential represents the higher ORR activity. The overpotentials of MnN_4_, MnN_4_−F, MnN_4_−Cl, MnN_4_−Br, and MnN_4_−I are 0.89, 0.64, 0.44, 0.41, and 0.39 V, respectively. Evidently, MnN_4_−I with the lowest overpotential of 0.39 V exhibits the best catalytic activity. Compared with the overpotentials of Pt (100) (0.47 V) [27] and Pt (111) (0.43 V) [28], MnN_4_−Cl and MnN_4_−Br/I, respectively, show comparable and higher catalytic activity, indicating that MnN_4_−Cl, MnN_4_−Br, and MnN_4_−I should be promising alternatives to Pt catalysts. Thus, X ligand modification is an effective strategy to improve the ORR catalytic activity of MnN_4_. Besides the 4e^−^ reaction pathway, the 2e^−^ reaction pathway (Equations (5) and (6)) [37,38] is also investigated to identify the ORR selectivity of all catalysts. The optimized adsorption structures of H_2_O_2_ on MnN_4_ and MnN_4_−X in Appendix A indicate that the H_2_O_2_ molecules cannot be stably adsorbed on MnN_4_, MnN_4_−Cl, and MnN_4_−Br and decompose into *O and H_2_O, which is consistent with the hydrogenation products of *OOH in the 4e^−^ reaction pathway, implying the better 4e^−^ reaction pathway selectivity. Although MnN_4_−F and MnN_4_−I can physically adsorb H_2_O_2_ with the adsorption energies of −0.08 and −0.06 eV, the free energy changes of *OOH to *O+H_2_O at zero electrode potential for MnN_4_−F (−1.70 eV) and MnN_4_−I (−1.86 eV) are significantly smaller than that (−0.20 and −0.39 eV) for *OOH to H_2_O_2_, indicating the 4e^−^ reaction pathway is easier to occur. Therefore, the ORR tends to follow the 4e^−^ reaction pathway on all catalysts.

### 2.4. Origin of the Catalytic Activity

To specifically analyze the origin of excellent catalytic activity of MnN_4_−X, the projected density of states (PDOS), *d*-band center, and the local density of states (LDOS) of Mn in MnN_4_ and MnN_4_−X were presented in Figure 6. It can be seen from this figure that that the *d*_yz_ and *d*_xz_ orbitals of Mn atoms in MnN_4_ are mainly located near the Fermi-level. When incorporating X ligand into MnN_4_, the *d*-orbital of the Mn atom moves towards the Fermi-level, and the main contribution near the Fermi-level is transformed into the $d_{z^{2}}$ orbital, resulting in the enhanced reaction activity. Therefore, the O_2_ adsorption configuration is transformed from side-on for MnN_4_ into end-on for MnN_4_−X. Moreover, the $d_{z^{2}}$ orbital occupations near the Fermi-level are significantly weaker than *d*_yz_ and *d*_xz_ orbitals in MnN_4_, and the *d*-band center of the Mn atom in MnN_4_ downshift from −0.81 eV to −1.10, −1.13, −1.11, and −1.07 eV in MnN_4_−F, MnN_4_−Cl, MnN_4_−Br, and MnN_4_−I, respectively. These changes result in the weakened adsorption capacity of MnN_4_−X for reaction intermediates as compared with MnN_4_. In particular, MnN_4_−Cl, MnN_4_−Br, and MnN_4_−I present appropriate adsorption strength for reaction intermediates and excellent ORR catalytic activity. In addition, the electronic band structures play an important role in determining the electronic transmission properties of catalysts [39]. Hence, we further examine the total density of states (TDOS) of MnN_4_ and MnN_4_−X. As shown in Appendix A, after modifying with X, both spin-up and spin-down orbitals of MnN_4_ cross through the Fermi-level, achieving the transition from semiconductor to metallic behaviors. This is conducive to the electron transfer during the reaction process, thereby enhancing the ORR catalytic activity. Therefore, halogen ligand modification can be an effective strategy to regulate the electronic structure of metal active center and improve the ORR catalytic activity of graphene-based SACs.

## 3. Computational Method

All spin-polarized DFT calculations were performed using the Vienna ab initio Simulation Package (VASP) [40]. The Perdew–Burke–Ernzerhof (PBE) within the generalized gradient approximate functional (GGA) was used to deal with exchange–correlation interactions [40,41]. For all calculations, the kinetic energy cutoff was chosen to be 500 eV, thus achieving good energy convergence. The thickness of the vacuum layer was set to be 15 Å to avoid interactions between mirror images. All structures were allowed to converge until Feynman Hallman forces were smaller than 0.02 eV/Å, and the total energy fluctuation was set to be 10^−5^ eV. The Brillouin zone was sampled with 5 × 5 × 1 *k*-points for structural optimization, and 11 × 11 × 1 *k*-points were sampled for the electronic structure calculations. The DFT-D3 method with the Becke–Johnson damping function was used to correct the van der Waals force dispersion [42]. Ab initio molecular dynamics (AIMD) simulations were performed using the Nosé–Hoover thermostat at 300 K in the canonical ensemble (NVT) with a time step of 1 fs and a total time scale of 10 ps.

ORR includes two different competing reaction pathways, namely, the 4e^−^ reaction pathway for the reduction of O_2_ to H_2_O and the 2e^−^ reaction pathway for the reduction of O_2_ to H_2_O_2_.The 4e^−^ reaction process under an acidic medium is listed as follows
(1)$$*+O_{2}+H^{+}+e^{-}=*\mathrm{OOH}$$
(2)$$*\mathrm{OOH}+H^{+}+e^{-}=*O{+H}_{2}O$$
(3)$$*O+H^{+}+e^{-}=*\mathrm{OH}$$
(4)$$*\mathrm{OH}+H^{+}+e^{-}=H_{2}O$$

The 2e^−^ reaction process under an acidic medium is presented as follows
(5)$$*+O_{2}+H^{+}+e^{-}=*\mathrm{OOH}$$
(6)$$*\mathrm{OOH}+H^{+}+e^{-}=H_{2}O_{2}$$

The change in Gibbs free energy for each step of the reaction is defined as
(7)$$\Delta G=\Delta E_{\mathrm{DFT}}+\Delta E_{\mathrm{ZPE}}-T\Delta S+\Delta G_{U}+G_{\mathrm{pH}}$$
where $\Delta E_{\mathrm{DFT}}$ represents the total energy calculated by DFT. $\Delta E_{\mathrm{ZPE}}$ and ΔS, respectively, are the changes in zero-point energy and entropy. T stands for the room temperature of 298.15 K.$\;\Delta G_{U}=-\mathrm{neU}$*,* where *n* is the number of electrons required to complete the reaction, and *U* is the electrode potential.$\;\Delta G_{\mathrm{PH}}=k_{B}T\times \ln10\times \mathrm{pH}$, in which $k_{B}$ stands for the Boltzmann’s constant, and pH is set to 0 in an acidic medium.

For the 4e^−^ reaction pathway, the step with the maximum free energy barrier is considered to be the RDS. The overpotential of the whole reaction progress is calculated by the following equation
(8)$$\eta =1.23+\max(\Delta G_{1},\;\Delta G_{2},\;\Delta G_{3},\;\Delta G_{4})/e$$

The binding energy (*E*_b1_) between Mn and N−coordinated graphene (N_4_−G) and the binding energy (*E*_b2_) between MnN_4_ and X are, respectively, given by the following equations
(9)$$E_{b1}=E_{\mathrm{Mn}}+E_{N_{4}-G}-E_{\mathrm{Mn}N_{4}}$$
(10)$$E_{b2}={\;E}_{\mathrm{Mn}N_{4}}+{1/2E}_{X_{2}\;}-E_{\mathrm{Mn}N_{4}-X}$$
where $E_{\mathrm{Mn}}$,${\;E}_{N_{4}-G}$, $E_{\mathrm{Mn}N_{4}}$, and $E_{\mathrm{Mn}N_{4}-X}$ represent the total energies of a single Mn atom, N_4_−G, MnN_4_, and X modified MnN_4_, respectively. $E_{X_{2}}$ represents the energy of a single halogen molecule.

The formation energies (*E*_f1_ and *E*_f2_) of MnN_4_ and MnN_4_−X are, respectively, defined as
(11)$$E_{f1}={\;E}_{\mathrm{Mn}N_{4}\;}-{\;E}_{\mathrm{Mn}}-\;4\mu_{N}-\;44\mu_{C}$$
(12)$$E_{f2}=E_{\mathrm{Mn}N_{4}-X}-E_{\mathrm{Mn}}-\;4\mu_{N}-44\mu_{C}-\mu_{X}$$
where $\mu_{C}$, $\mu_{N}$, and $\mu_{X}$ denote the chemical potentials of C, N, and halogen atoms, which can be obtained from perfect graphene, nitrogen gas, and halogen molecules, respectively.

The adsorption energy (*E*_ads_) of catalysts for reaction intermediates is defined as the following equation
*E_ads_* = *E_sup+inter_* − *E_sup_* − *E_inter_*(13)
where *E_sup+inter_*, *E_sup_*, and *E_inter_* represent the energies of the adsorption system, the individual support, and the reaction intermediates, respectively.

In addition, the electrochemical stability of catalysts can be evaluated by the solution potential (*U*_diss_) [43], which is defined as
(14)Udiss=Udiss°(metal, bulk)−Efne
where $U_{\mathrm{diss}}^{{}^{\circ}}\left(\mathrm{metal},\;\mathrm{bulk}\right)$ and n represent the standard dissolution potential of bulk metal and the number of electrons involved in the dissolution, respectively. *U*_diss_ > 0 vs. the standard hydrogen electrode means that the system is electrochemically stable.

## 4. Conclusions

In summary, the structural stabilities, electronic structures, and ORR catalytic performances of MnN_4_ with and without halogen ligand X (X = F, Cl, Br, and I) modification were systemically investigated by DFT calculations. The calculated results show that the MnN_4_ and MnN_4_−X catalysts are thermodynamically and electrochemically stable, and follow the four-electron ORR pathway. After introducing axial X into MnN_4_, the X can obtain some electrons from the Mn atom in MnN_4_ and regulate the electronic structures of the Mn atom. The *d*_yz_ and *d*_xz_ orbitals of the Mn atom near the Fermi-level are transformed into the weaker $d_{z^{2}}$ orbital, and the *d*-band center of the Mn atom is shifted away from the Fermi-level, resulting in the reduced adsorption strength of MnN_4_ for *OH. Thus, all MnN_4_−X catalysts present enhanced ORR catalytic activity compared with MnN_4_. In particular, MnN_4_−Br and MnN_4_−I exhibit excellent ORR catalytic activity, respectively, with the overpotentials of 0.41 and 0.39 V, which are superior to MnN_4_ and Pt catalysts. Halogen ligand modification was identified to be an effective strategy for improving the ORR catalytic activity. This study is expected to provide a new perspective for the design and development of highly active graphene-based single-atom ORR catalysts.

## Figures and Tables

**Figure 1 molecules-29-03517-f001:**
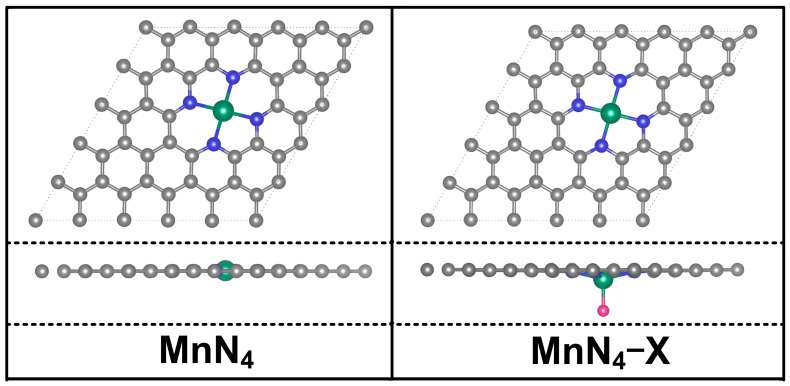
Schematic diagram of MnN_4_ and MnN_4_−X structures. Grey, blue, green, and purple balls represent C, N, Mn, and X (X = F, Cl, Br, and I) atoms, respectively.

**Figure 2 molecules-29-03517-f002:**
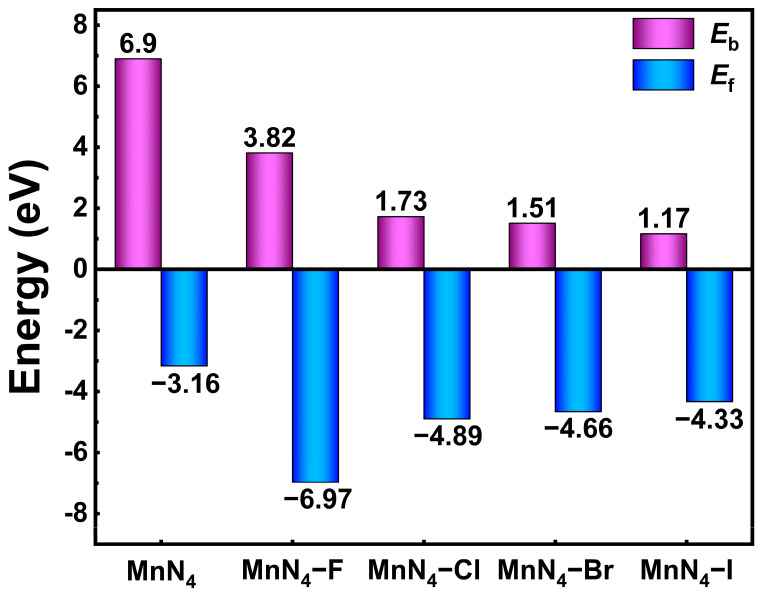
The binding and formation energies for MnN_4_ and MnN_4_−X.

**Figure 3 molecules-29-03517-f003:**
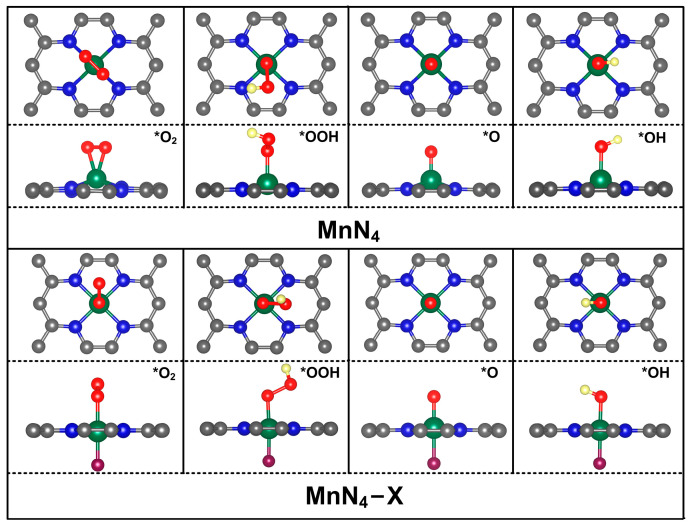
Schematic diagram of the most stable adsorption structures of intermediates on MnN_4_ and MnN_4_−X. Grey, blue, green, purple, red, and yellow balls represent C, N, Mn, X, O, and H atoms, respectively. “*” indicates the clean surface.

**Figure 4 molecules-29-03517-f004:**
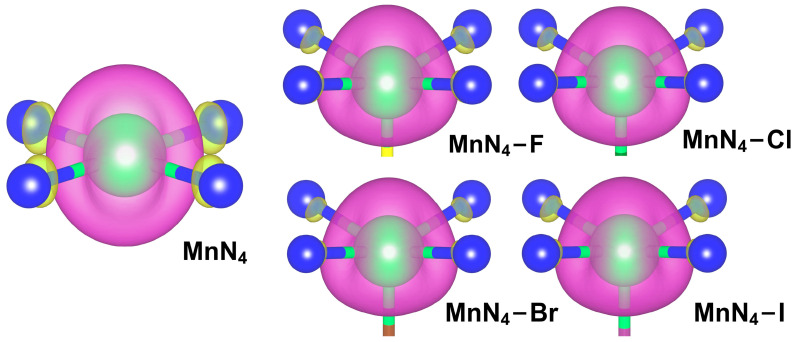
Spin-resolved charge densities of Mn atom in MnN_4_ and MnN_4_−X with an isosurface value of 0.06 e/Å^3^. Green and blue balls represent Mn and N atom, respectively.

**Figure 5 molecules-29-03517-f005:**
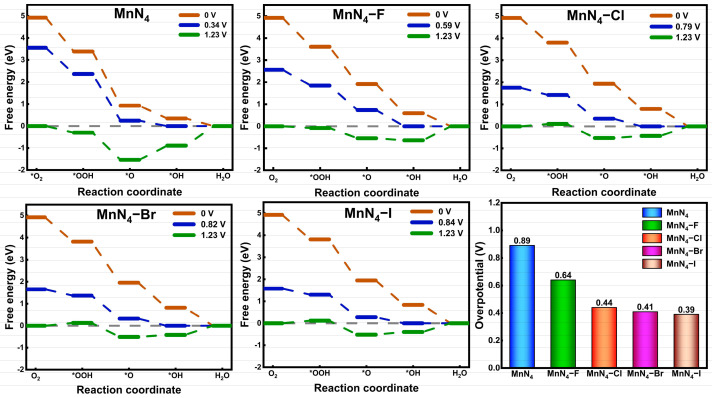
Free-energy diagrams and overpotentials of ORR on all catalysts. “*” indicates the clean surface.

**Figure 6 molecules-29-03517-f006:**
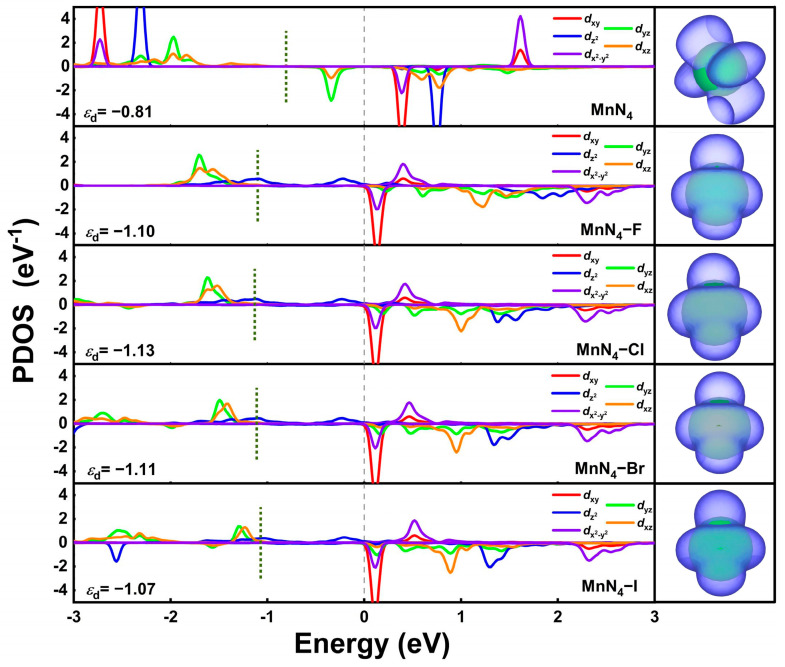
The PDOS and the LDOS within the range of −0.6 to 0 eV for Mn in MnN_4_ and MnN_4_−X.

## Data Availability

Data are contained within the article and Appendix A.

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
