# Peer review of "Boosting the Electrocatalytic Oxygen Reduction Activity of MnN4-Doped Graphene by Axial Halogen Ligand Modification"

_molecules, 2024, doi:10.3390/molecules29153517_

Round 1

Reviewer 1 Report

Comments and Suggestions for Authors

The authors calculated MnN4-doped graphene by axial halogen ligand modification for efficient ORR. Some issues should be well solved before acceptance.

1.     The mistakes about superscripts and subscripts should be corrected through the manuscript, such as dz2.

2.     Reliable reference for the analysis of bader charge should be provided. Please refer to 10.1063/5.0083059.

3.     Band gap is a very important orbital property for analyzing catalysis electronic transport, which should be further analyzed by referring to 10.1038/s41467-019-11847-w for this point.

4.     Some necessary literatures should be added to support the computational results in this work.

Comments on the Quality of English Language

Minor editing of English language required

Reviewer 2 Report

Comments and Suggestions for Authors

This paper presents a computational study on the effect of axially coordinated halides on the catalytical activity of MnN4 systems in the electrocatalytic reduction of oxygen. The work is interesting and well structured, and the reported results provides useful insights on the topic, therefore I think this paper is worthy of publication in Molecules. Nevertheless, I believe the following revisions would be beneficial for the paper before its acceptance.

1) The introduction structure is effective and the references are updated. However, it is quite difficult to understand which state of the art results are only computational and which ones are corroborated by experimental data or purely experimental. This aspect should be made clearer throughout the introduction to make it more sounding and explanatory. When available, it could also be interesting to report a comparison between experimental and computational data.

2) From lines 50-53, it is clear that the computational study of MnN4 -X systems is a novelty from this work. On the other hand, it is not clear if the computational study of MnN4 is also novel. If a study on the MnN4 system has already been reported, I would recommend more comparisons with literature data in the results and discussion section.   

3) Lines 211-218. The sentence "Because the H2O2 molecule cannot be formed on MnN4, MnN4-Cl, and MnN4-Br (see Fig. S2), the 2e- reaction pathway does not exist." should be modified and better integrated with the other sentences in this group of lines, where the predominance of the 4e- pathway is explained.

4) Lines 245-249. The sentence "The incorporated halogen ligands..." could be rephrased in a clearer way. 

Comments on the Quality of English Language

The English of the manuscript is overall of good quality and clear.

Round 2

Reviewer 2 Report

Comments and Suggestions for Authors

The paper has been improved by satisfactorily implementing the requested revisions. I recommend the publication of the paper in present form.